# The Usefulness of Thyroid Antibodies in the Diagnostic Approach to Autoimmune Thyroid Disease

**DOI:** 10.3390/antib12030048

**Published:** 2023-07-22

**Authors:** Hernando Vargas-Uricoechea, Juan Patricio Nogueira, María V. Pinzón-Fernández, Diego Schwarzstein

**Affiliations:** 1Metabolic Diseases Study Group, Department of Internal Medicine, Universidad del Cauca, Carrera 6 N° 13N-50, Popayan 190001, Colombia; 2Centro de Investigación en Endocrinología, Nutrición y Metabolismo (CIENM), Facultad de Ciencias de la Salud, Universidad Nacional de Formosa, Formosa P3600AZS, Argentina; 3Health Research Group, Department of Internal Medicine, Universidad del Cauca, Popayan 190001, Colombia; mvpinzonf@gmail.com; 4Endocrinology Service, Servidigest Clinic, 08006 Barcelona, Spain; schw.diego@gmail.com

**Keywords:** thyroid, autoimmunity, antibodies, thyrotropin, receptor, thyroglobulin, peroxidase

## Abstract

Autoimmune thyroid disease (AITD) refers to a spectrum of various diseases, with two extremes of clinical presentation, hypothyroidism (Hashimoto’s thyroiditis (HT) and hyperthyroidism (Graves–Basedow disease (GBD)). Both conditions are characterized by presenting a cellular and humoral autoimmune reaction, with an increase in the synthesis and secretion of antibodies directed toward various thyroid antigens, together with a phenomenon of thyrocyte necrosis and apoptosis (in HT) and a persistent thyrotropin-receptor stimulation (in GBD). The diagnosis of both entities is based on clinical, laboratory, and imaging findings. Three major anti-thyroid antibodies have been described, those directed against the TSH receptor (TRAb), against thyroid peroxidase (TPOAb), and against thyroglobulin (TgAb). Each of these autoantibodies plays a fundamental role in the diagnostic approach of autoimmune thyroid disease. TRAbs are the hallmark of GBD, and additionally, they are predictors of response to disease treatment, among other utilities. Likewise, TPOAb and TgAb allow for identifying individuals with a higher risk of progression to hypothyroidism; the positivity of one or both autoantibodies defines the presence of thyroid autoimmunity. In this review, the usefulness of anti-thyroid antibodies in the diagnostic approach to autoimmune thyroid disease is described.

## 1. Introduction

Immune tolerance is defined as a lack of response to an antigen, induced by previous exposure to said antigen; therefore, tolerance to self-antigens is a fundamental property of a “normal” immune system. Loss of tolerance to self-antigens leads to an inappropriate immune reaction called autoimmunity [1,2].

Diseases caused by such reactions are called autoimmune diseases (AIDs) and are characterized by pathogenic inflammatory responses induced by T lymphocytes (TL) and B lymphocytes (BL), which can induce “autotoxic” effects in virtually any organ or system [3,4].

When there is multi-organ involvement, the AID is classified as non-organ-specific (as in systemic lupus erythematosus and rheumatoid arthritis, among others), and when it affects a specific organ, it is classified as an organ-specific AID (such as type 1 diabetes, pernicious anemia, and autoimmune thyroid diseases (AITD), among others) [5,6].

Although the molecular mechanisms that induce AIDs are complex, it is clear that some genetic, non-genetic, epigenetic, and environmental factors are the basis for explaining the pathogenesis of these diseases and predicting clinical and biochemical responses [7,8].

AITD is the most common AID globally, presenting two classic phenotypes, hypothyroidism (subclinical or primary) in the context of Hashimoto’s thyroiditis (HT), or hyperthyroidism (subclinical or primary) in the context of Graves–Basedow disease (GBD) [9,10].

However, there are other thyroid conditions within the AITD group, such as postpartum thyroiditis, thyroiditis associated with autoimmune polyglandular syndromes, and drug-induced thyroiditis (for example, amiodarone), among others [11,12].

AITD is characterized by lymphocytic infiltration of the thyroid gland. For instance, in HT, the consequent inflammation induces follicular cell destruction, necrosis, and apoptosis, with subsequent fibrosis (and potentially hypothyroidism), with a humoral antibody (Ab)-mediated response directed against one or several thyroid antigens. These can include thyroid peroxidase (TPO) and thyroglobulin (Tg), among others [13,14].

In GBD, on the other hand, a humoral response predominates, with the presence of Abs that stimulate the thyrotropin (TSH) receptor (TSHR). It can be accompanied by goiter, hyperthyroidism, ophthalmopathy, and dermopathy (Figure 1) [15,16].

In this review, the three Abs directed against the major thyroid antigens (TPO, Tg, and TSHR) and against other minor thyroid antigens (pendrin (PDN), sodium iodide symporter (NIS), and megalin (Meg)) are described, as well as the prevalence of the positivity of said antibodies in individuals with AITD and their usefulness in clinical practice.

## 2. Methods (Search Strategy)

We performed a detailed search in the following databases: PubMed; PubMed Central; Scopus; EMBASE; BIOSIS; UpToDate; and Web of Science. Articles were selected according to the following keywords: autoimmune thyroiditis; Hashimoto’s thyroiditis; Graves–Basedow disease; thyroid antibodies; thyroid antigens; and autoimmune thyroid disease. Only English-written articles were included (Figure 2).

## 3. Major Thyroid Antigens

### 3.1. Tg

The gene encoding Tg synthesis is a single copy gene (270 kb in length) located on chromosome 8q24.2–8q24.3, which contains an 8.5 kb coding sequence divided into 48 exons [17].

Tg is a glycosylated protein synthesized exclusively in the thyroid and is the largest (660 kDa) and most abundant autoantigen of the thyroid. It is essential for the synthesis of both thyroid hormones (TH) T4 and T3 since the synthesis of these hormones depends on the conformation, iodination, and post-translational modification of Tg [18].

TH synthesis (from Tg) occurs in the thyroid via iodination and coupling of pairs of tyrosines, which is completed by Tg proteolysis. Additionally, follicular Tg is capable of suppressing the thyroid feedback phenomenon since it can inhibit the expression of TTF-1, TTF-2, and PAX-8, decreasing the expression of the genes that code for the synthesis of TPO, NIS, TSHR, and Tg [18,19].

This suggests that Tg is not only the substrate for the synthesis of TH but also a regulator of thyroid function [18,19].

### 3.2. TPO

The gene encoding TPO synthesis is a single copy gene (2pter—p12) that codes for a protein of 933 amino acids, spanning the cell membrane, with a large extracellular domain (with 848 amino acids and five potential glycosylation sites) facing the follicular lumen and a cytoplasmic tail with a length of 61 amino acids [20,21].

The extracellular domain consists of three regions, which denote a high degree of sequence similarity, with other domains of specific three-dimensional structure, such as the myeloperoxidase-like domain, the complement control protein-like domain, and the epidermal growth factor-like domain. TPO expression is controlled by transcription factors, such as TTF-1, TTF-2, and PAX-8 [22,23].

TPO has two active sites, which facilitate the iodination of tyrosine residues in Tg, together with the dual oxidase enzyme and hydrogen peroxide, which subsequently allows for intrachain coupling of two iodotyrosine residues for TH synthesis. In addition, TPO catalyzes two reactions within the thyroid, the oxidation of iodine and the coupling of iodinated tyrosines in the process of TH synthesis [23,24].

Therefore, TPO plays a key role in the biosynthesis of TH and is a fundamental component of normal thyroid function.

### 3.3. TSHR

TSHR is encoded by a gene located at chromosome 14q31 and belongs to the family of G-protein-coupled receptors. It has a large extracellular domain (containing an N-terminal domain, a leucine-rich repeat domain, and a hinge region or cleavage domain), seven transmembrane passageways, and a small intracellular domain [25,26]).

TSHR couples to four G protein subfamilies, including Gs (inducing adenylyl cyclase activity and cAMP production), phospholipase C (which activates Gq/G11), G13 (which, in turn, is capable of inducing p44/42 mitogen-activated protein kinase), and Gi (which inhibits adenylyl cyclase activity) [27,28].

Mature TSHR contains two subunits (A and B). The A subunit is made up of a large extracellular domain mainly determined by multiple leucine-rich repeats and an N-terminal tail, where specific amino acids fold to form a complex TSH-binding pocket. The B subunit contains a short portion anchored to the membrane and the intracellular portion of the receptor [29,30].

TSHR expression occurs mainly on the basolateral membrane of thyrocytes, and its activation stimulates iodine uptake, TH synthesis and secretion, and thyrocyte proliferation [30,31].

However, TSHR expression has also been demonstrated in other tissues, such as the hypothalamus, cerebellum, amygdala, cortex, cingulate gyrus, frontal, occipital and temporal lobes, periorbital tissue, epidermis, hair follicles, kidneys, ovaries, testicles, adrenals, bone marrow hematopoietic cells, thymocytes, and antigen-presenting cells (APCs), among others [31,32].

In the thyroid, TSHR activation and G protein binding induce a series of complex intracellular events, which, consequently, stimulate the growth of thyroid cells and the production of TH [33].

## 4. Minor Thyroid Antigens

### 4.1. NIS

The gene that codes for the synthesis of NIS is located on chromosome 19p12–13.2 and encodes a glycoprotein of 643 amino acids with a molecular mass of about 70–90 kDa [34].

NIS is an intrinsic membrane protein belonging to the superfamily of sodium/solute symporters and the human transporter family (SLC5), containing 13 transmembrane domains, an extracellular amino terminus, and an intracellular carboxy terminus [34,35].

In thyrocytes, NIS is located at the basal cell level and is a mediator of active iodine transport to thyroid follicular cells involved in the first step of TH synthesis [36].

Iodine transport, mediated by NIS, is a vector process stimulated by TSH; NIS is also expressed in other tissues, such as the salivary glands, ductal cells, placenta, testicular cells, stomach, and mammary glands (during lactation). However, our understanding of the physiological role of the symporter in these tissues is not yet conclusive [36,37].

This symporter is capable of co-transporting two sodium ions with one iodide ion, and the resulting sodium gradient across the membrane acts as a driving force for iodide uptake. Furthermore, TSH is considered to be the factor that regulates NIS expression; in fact, TSH can stimulate iodide uptake by increasing NIS transcription (through cAMP) [37,38].

Additionally, TSH can also mediate this phenomenon through post-transcriptional mechanisms. Therefore, NIS catalyzes the accumulation of iodide in thyroid cells, an essential step in TH synthesis [39].

### 4.2. PDN

The gene encoding PDN synthesis is the same gene responsible for Pendred syndrome, which is an autosomal recessive disease that manifests with goiter and congenital sensorineural deafness. PDN is encoded by the SLC26A4 gene, which is located on chromosome 7q21-31 and contains 21 exons with an open reading frame of 2343 bp [40,41].

PDN (SLC26A4) is a glycoprotein composed of 780 amino acids. It contains three putative extracellular asparagine glycosylation sites and is considered an apical membrane-bound iodide transporter that acts as a multifunctional exchanger of multiple monovalent anions (for example, iodide, chloride, and bicarbonate) and is highly expressed both in the thyroid and in extrathyroid tissues (inner ear and kidneys). In the thyroid, PDN is expressed in the apical membrane of thyrocytes and participates in the transport of iodide to the colloid, indicating its importance in the TH synthesis [42,43,44].

### 4.3. Meg

Meg is a giant 600 kDa cell surface protein. Its gene covers 235,000 base pairs on the human chromosome 2q24-q31 and consists of 79 exons. It belongs to the endocytic low-density lipoprotein receptor family, which is expressed on the apical surface of thyrocytes; this expression is mediated by TSH. Meg has a high binding affinity for Tg and allows (at least in part) its uptake by thyrocytes; once Tg is internalized by Meg, lysosomal metabolism is avoided, and it is able to reach the basolateral membrane of the thyrocytes (via transcytosis), from where it is released into the blood [45,46,47].

During this transcytosis process, a portion of Meg remains complexed with Tg and enters circulation, so it could be considered an autoantigen that eventually leads to an Ab-mediated response. However, despite the fact that about half of patients with AITD may present MegAbs, its role in the pathogenesis and usefulness in the diagnosis, management, and follow-up of AITD is unknown [48,49].

The molecular characteristics and the prevalence of positivity of the different thyroid Abs are summarized in Table 1.

## 5. Major Thyroid Abs

### 5.1. TgAbs

The ability of Tg to induce an immune response depends, at least in part, on the content of both T4 and T3. The concentration of TH within Tg is capable of changing its conformation, stimulating the formation of masked and unmasked epitopes. Consequently, the binding capacity of the Abs can be affected by the content of T4 and T3 in Tg [51].

TgAbs identified in individuals without AITD generally recognize highly conserved epitopes (located in the T4 and T3-containing regions of Tg), whereas, in individuals with AITD, TgAbs are less restricted. Additionally, TgAbs mainly recognize “conformational” and, to a lesser extent, “linear” epitopes, suggesting that the immunogenic potential of Tg increases to the extent that its fragments have a greater capacity to generate conformational epitopes [50,52].

In general, TgAbs are polyclonal (of the IgG class), with different contributions according to the four subclasses (IgG4 > IgG3 > IgG2 > IgG1), although low levels of IgA, kappa, and lambda light chains have also been described [53].

Likewise, this humoral response is highly restricted to two immunodominant regions of Tg (143, 144, 147, and 150–154). In fact, TgAbs are responsive toward restricted epitopes located mainly in the central region and at the C-terminus of Tg (144, 153, and 155–159) [54].

TgAb formation can be generated by the massive release of Tg after tissue destruction (as a consequence of thyroiditis, by direct tissue trauma, or by manipulation of the thyroid), although the intrathyroidal iodine content must also be taken into account. For example, excessive iodine consumption can change the conformation of Tg, making it much more antigenic [55,56].

### 5.2. TPOAbs

TPOAbs are considered the hallmark of AITD. The prevalence of positivity in individuals with AITD is higher than for TgAbs [57].

TPOAbs are capable of recognizing discontinuous determinants in TPO, which have been named immunodominant regions A (IDR-A) and B (IDR-B), and several contact residues constituting IDR-A and IDR-B have been identified: 225; 353–363; 377–386; 597–604; 611–618; 620; 624; 627; 630; 646; 707; 713–720; and 766–775 [58,59].

TPOAbs can also react against conformational or linear epitopes, and the polyclonal Abs present both in healthy individuals and those with AITD are directed against the same epitopes, taking into account that TPOAbs from healthy individuals do not block TPO action, while those identified in AITD patients can fix complement, produce lysis of thyrocytes, and competitively inhibit enzymatic activity [60].

Additionally, they can induce oxidative stress. However, despite the cytotoxic effect of TPOAbs on HT, their role in individuals with GBD has not been fully established. It has also been found that the spatial arrangement of the epitopes, together with the domain architecture and the positioning in the membrane, suggest that the interaction of TPO with TPOAbs may require radical changes in the tertiary structure of the antigen [61,62].

TPOAbs can be of any IgG class, although the estimated prevalence in descending order is the following: IgG1 (70%); IgG4 (66.1%); IgG2 (35.1%); and IgG3 (19.6%). Low levels of IgA-type TPOAbs have also been described in some patients [63].

The most frequently used laboratory methods in the evaluation of TgAbs and TPOAbs, in addition to the prevalence of positivity in patients with AITD and in healthy individuals, are summarized in Table 2 and Table 3.

### 5.3. TRAbs

Similar to TPOAbs being considered the hallmark of HT, TRAbs are the hallmark of GBD. The prevalence of TRAbs in subjects with HT is 10–20%, and in GBD, it is 90–95%; hence, its detection is recommended in the differential diagnosis of patients with hyperthyroidism [71].

The mechanism by which hyperthyroidism occurs in GBD is due to the presence of TRAbs, which simulate the effects of TSH on thyrocytes. TSHR is a receptor that belongs to the 7TM G-protein-coupled receptor family and is expressed in thyroid follicular cells (and also in thymocytes and retroorbital tissue fibroblasts) [72,73].

From the functional and biological points of view, TRAbs can be classified in three ways, stimulators, blockers, and neutral; for GBD, the most frequent are the stimulators. Stimulator TRAbs bind to the N-terminus of the TSH extracellular domain and consequently stimulate TH production (independently of the feedback phenomenon of the hypothalamic–pituitary–thyroid axis) [74,75].

TRAbs have a high receptor affinity; however, their absolute concentration is low. One explanation for this may be that they are produced by a limited number of BLs and APCs. Moreover, in some individuals, the immune response may alternate and change from a state, in which stimulatory TRAbs are initially (and predominantly) produced, to an opposite state, in which the production of blocking or neutral TRAbs is increased, resulting in changes in the clinical and biochemical findings of the disease [76,77].

TRAbs are a combination of highly related IgGs, which have the ability to bind to specific epitopes of the TSHR. However, these Abs can vary and fluctuate within the same individual (and between individuals). Therefore, the presence of subtle changes in the affinity or specificity of TRAbs can cause radical changes in their ability to activate the TSHR [77,78].

According to the detection methods commonly used for the measurement of TRAbs, these can be divided into competitive immunoassays, bioassays, and enzyme-linked immunosorbent assay (ELISA) [79].

The former can detect all types of TRAbs, determining their ability to compete with a labeled ligand (TSH or a monoclonal antibody against TSHR) and their ability to bind to TSHR. Furthermore, bioassays can measure the stimulatory or blocking effect of TRAbs by detecting cAMP production and the intracellular TSHR signal (via TSHR-expressing cells). Finally, the ELISA method is based on the inhibition of human monoclonal TRAb (M22) binding [71,79,80].

The measurement methods and precision of the most commonly used TRAbs in the diagnostic approach of GBD are described in Table 4.

### 5.4. NISAbs and PDNAbs

The prevalence of positivity for NISAbs and PDNAbs in individuals with AITD and in the general population is highly variable. In general terms, the prevalence of NISAb positivity in healthy individuals is very low; however, its prevalence is increased in those with AITD (especially in GBD subjects) [85].

Moreover, some studies have documented that the prevalence of positivity for NISAbs and PDNAbs is similar in patients with AITD (prevalence close to 10% for each Ab). Likewise, the prevalence of PDNAbs positivity is only slightly higher in individuals with AITD (relative to healthy controls, being also detectable in the latter), and the prevalence is higher in individuals with GBD (versus HT and participants without AITD) [85,86].

Several explanations for these findings can be given, such as the type of population studied (populations with low prevalence of the disease), studies with small sample sizes, and the type of technology used to measure Abs. For these reasons, the role of NISAbs and PDNAbs in the diagnosis, prediction, and response and relapse rate of AITD still needs to be clarified.

The measurement methods for NISAbs and PDNAbs most frequently used in the evaluation of individuals with AITD are summarized in Table 5 and Table 6. Likewise, Table 7 summarizes the prevalence of positivity for NISAbs and PDNAbs in subjects with GBD, HT, and in participants without AITD in different studies.

### 5.5. MegAbs

As previously noted, Meg transports Tg through the thyroid epithelial cells, subsequently entering circulation in a Tg–Meg complex; thus, Meg is capable of eliciting an Ab-mediated immune response [105].

In rodent models of Heymann’s nephritis, Meg has the ability to induce Ab production and secretion (Heymann’s nephritis is an experimental rat model for active and passive immune-mediated nephritis). However, Meg, which is the target antigen, is localized in podocytes in the rat model, but in humans, megalin is found in the proximal tubule and not in podocytes [106,107].

This experimental model in rodents allowed us to propose that MegAbs could be generated and manifest in individuals with AITD in the same way. Initially, studies measuring IgG binding to L2 cells (a rat yolk sac carcinoma cell line known to express Meg) found a prevalence of 50% in subjects with HT and a lower percentage in individuals with GBD (10.5%), while it was not present in healthy individuals [108].

These results led to theories of a possible role of MegAbs in AITD being disconfirmed, although, in fact, there are very few studies evaluating the usefulness of MegAbs in different clinical and/or biochemical outcomes in individuals with AITD.

Some of the methods used for the detection of MegAbs are summarized in Table 8.

## 6. Clinical Utility of Thyroid Abs in AITD

In AITD, the high prevalence of thyroid Abs (especially TRAbs, TPOAbs, and TgAbs) has made it possible to assess their usefulness in the initial approach and in the follow-up of these patients.

These Abs have been evaluated in aspects, such as diagnosis (their mere presence determines the diagnosis of thyroid autoimmunity), differential diagnosis (for example, in cases where the symptoms and images do not allow for differentiation between “Hashitoxicosis” and GBD), treatment (making it possible to predict, to a certain extent, which of the individuals affected with GBD may respond better to treatment with antithyroid drugs (ATD)), risk of relapse (in people affected by GBD with strongly positive TRAbs, the risk of relapse or recurrence of the disease is higher), and prognosis (in individuals with any of the major antibody positives, there is an increased risk of developing hypothyroidism or hyperthyroidism over time, depending on the type of Ab or Abs present) [111].

### 6.1. Clinical Utility of TgAbs

Routine measurement of TgAbs in iodine-“sufficient” areas does not seem to be very useful as a screening method in the study of AITD; however, it may be useful in iodine-“deficient” areas, particularly in individuals with nodular goiter [112].

It has also been found that, in geographic regions where universal salt iodization programs (for consumption) have been developed, there has been a significant increase in the positivity of TgAbs after salt iodization, indicating that an eventual excess consumption of iodine (through salt) may also increase the immunogenicity of Tg, leading to a higher rate of thyroid autoimmunity. Therefore, it could be argued that in areas where programs of salt iodization have been implemented and where excess consumption has been documented, the measurement of TgAbs could play an important role in the population characterization of thyroid autoimmunity and the potential risk of developing thyroid functional disorders [9,63,113].

### 6.2. Clinical Utility of TPOAbs

In patients with hypothyroidism (subclinical or primary), TPOAb positivity determines the diagnosis of HT, while in euthyroid individuals, TPOAb positivity is associated with a significant increase in the risk of developing hypothyroidism over time [114,115].

Likewise, TPOAbs have been associated with the development of ocular alterations in patients with GBD. However, the results have been contradictory, especially in children. These findings have not been corroborated on a large scale in adults [14,116].

Moreover, TPOAbs have been related to a higher rate of infertility, premature birth, and spontaneous abortions (independent of TSH or T4 levels). In fact, it is recommended to evaluate TPOAbs levels in the pre-pregnancy period and once a pregnancy is confirmed since their presence (even with TSH values in the normal range of 2.5–4.0 mIU/L) could indicate the need for levothyroxine replacement [117].

In addition, the presence of TPOAbs during early pregnancy may be associated (in children) with a lower intelligence quotient, although it seems that this is mediated by population iodine status, taking into account that TPOAbs can cross the placenta (based on the levels of TPOAbs in umbilical cord blood at the time of birth being similar to those of the mother in the last trimester of pregnancy). This finding does not seem to be related to changes in fetal thyroid function [116,117].

Furthermore, TPOAbs measurement during pregnancy may predict the risk of postpartum thyroiditis (with a higher predictive value when measured in the first trimester of pregnancy). There has also been increasing interest in the relationship between TPOAb positivity and the risk of hypothyroidism, destructive thyroiditis, and/or decreased thyroid volume in individuals receiving interferon-α, kinase inhibitor therapy, interleukin-2, amiodarone, and lithium [118].

### 6.3. Clinical Utility of TRAbs

The clinical utility of TRAbs in the diagnosis and monitoring of AITD remains controversial, despite the fact that their presence defines the diagnosis of GBD, especially in those individuals with long-standing hyperthyroidism associated with extrathyroid manifestations (ophthalmopathy, myxedema) [119].

Although in the diagnostic approach of hyperthyroidism (subclinical or primary), imaging methods such as ultrasound and/or thyroid scintigraphy can be used, the measurement of TRAbs can resolve the diagnosis in the majority of affected individuals, differentiating GBD from a “Hashitoxicosis” or other types of thyroiditis that debut with hyperthyroidism, or also fictitious thyrotoxicosis, and even toxic nodular goiter [120].

On the other hand, TRAbs can be detected in practically all patients with thyroid ophthalmopathy. In fact, TRAbs levels correlate with the severity and clinical activity of the disease, and in addition, a high level of TRAbs in patients with early ophthalmopathy predicts a poor prognosis. TRAbs are also useful in those individuals with clinical features of thyroid ophthalmopathy but with normal or discordant thyroid function (for example, with hypothyroidism) [121,122].

Likewise, elevated TRAbs prior to treatment with radioactive iodine (RAI) in individuals with GBD is associated with exacerbation of thyroid ophthalmopathy, suggesting that thyroidectomy could be considered (in the case of non-response to management with ATD) in this type of patients since in the long term, an increase in TRAbs levels has not been demonstrated in patients undergoing thyroidectomy [123].

The usefulness of measuring TRAbs in individuals with pretibial myxedema must be demonstrated. It may be useful in patients with findings suggestive of this entity but with normal thyroid function [124].

The detection of TRAbs can also help to make the differential diagnosis in individuals with amiodarone-induced thyroiditis (AmIT) since, in this scenario, the presence of TRAbs increases the positive predictive value of a type 1 AmIT; however, a minority of patients can coexist with an associated inflammatory component (type 2 AmIT), being a real challenge to be able to differentiate between both entities being inclusive in the management decision-making with ATD [125].

Additionally, it should also be taken into account that the absence of TRAbs does not rule out the diagnosis of AmIT; for this reason, the measurement of TRAbs, together with imaging studies (ultrasonography, scintigraphy) and evolution over time will allow confirmation of this diagnosis [126].

Measurement of TRAbs may also be useful in predicting the course of GBD since clinical remission of the disease is more likely to be achieved when a decrease in the level of TRAbs is documented (in individuals receiving treatment with ATD). In this sense, in patients with very high levels of TRAbs (together with prominent extraocular manifestations and large goiter), an “attenuated” and ineffective response can be predicted for the use of ATD [116,127].

Furthermore, after treatment with ATD, the recurrence rate of GBD is higher in those patients who initially had a decrease in TRAbs levels, but over time, their levels rose again, which suggests that TRAbs should be measured (in the long-term) in patients who have successfully completed ATD treatment [128].

Moreover, the presence of TRAbs predicts the risk of fetal/neonatal hyperthyroidism in the offspring of women with a history of AITD; in fact, 2–10% of pregnant women with very high levels of TRAbs have children with hyperthyroidism [129].

It should also be taken into account that the risk of fetal/neonatal hyperthyroidism is significantly reduced after ATD treatment in the mother but may be high in cases where RAI treatment was performed and TRAbs levels remained elevated. The measurement of TRAbs should be routine in these gestants (in the first and third trimesters) [129,130].

Likewise, the measurement of TRAbs in pregnant women receiving ATD (in the third trimester) is also recommended. The presence of TRAbs in these situations requires an exhaustive evaluation of hyperthyroidism in the fetus (throughout gestation) and in the neonate. In the latter, evaluation of complete thyroid tests and TRAbs should be performed at birth from the blood of the umbilical cord and, sequentially, until seven days postpartum (the time by which the transplacental passage of ATDs has disappeared) [130,131].

Moreover, and as previously noted, there are several types of TRAbs (in addition to those that stimulate TSHR (TSAb) and are involved in the pathogenesis of GBD) [13,14].

TRAbs that have the ability to block TSHr (TBAb) could have utility in the evaluation of subjects with autoimmune thyroiditis; however, the routine measurement of TRAbs (containing both TSAb and TBAb) is not recommended since they have a low prevalence in the global context of chronic autoimmune thyroiditis. Moreover, in those clinical settings where it is feasible to measure them, their presence does not change the treatment strategy [132,133,134,135].

However, there are some considerations or clinical situations where the detection of TBAb may be useful:a.In individuals with HT and hypothyroidism, where there is adequate and stable control with very low doses of levothyroxine, since it is possible in this type of patient that a release of TBAb has occurred, causing a type of transient hypothyroidism, which could have a high recovery rate;b.In a newborn born to a mother with HT but with the presence of contradictory or bizarre clinical findings;c.In patients with HT and clinical findings suggestive of thyroid ophthalmopathy;d.When in individuals with long-standing hypothyroidism, under treatment with stable doses of levothyroxine, a change to the state of hyperthyroidism is noted;e.When alternating periods of hyperthyroidism and hypothyroidism occur in the same patient.

Finally, the usefulness of TRAbs that have a “neutral” effect on thyroid function has not been demonstrated; therefore, their role in individuals with AITD is unknown (Figure 3) [133,134,135].

### 6.4. Clinical Utility of PDNAbs, NISAbs, MegAbs, and Other Abs

The utility of these Abs in the diagnosis, treatment, and prognosis of AITD remains to be demonstrated; to date, none have been shown to have superior diagnostic performance to that provided by TPOAbs, TgAbs, and TRAbs in individuals with HT and GBD [46,47,85,86,102,107].

Other Abs that have been described in patients with AITD are THAbs. THAbs are Abs that can bind to TH, and the presence of such Abs seems to be related to a massive “leakage” of Tg, exposing the immune system to several hormonogenic epitopes of the Tg and inducing a humoral (Ig-mediated) response. Four types of THAb have been described based on the presence of IgM or IgG (T4-IgM, T4-IgG, T3-IgM, and T3-IgG), with T4-IgG and T3-IgG being the most prevalent [136,137].

The occurrence of THAbs is variable; for example, in the general population, the prevalence is approximately 1%, but in individuals with AITD, the prevalence is 20–23% and 32–46% (in HT and GBD, respectively) [138,139].

Additionally, THAbs can also be found in the serum of patients with other autoimmune diseases, such as Sjögren’s syndrome and rheumatoid arthritis, suggesting the presence of epitopes that can induce a “cross-reaction” in other tissues, such as connective tissue. Moreover, a high frequency of THAbs has also been demonstrated in patients with type 1 diabetes mellitus or with vitiligo and in individuals with autoimmune polyglandular syndrome [140,141].

THAbs are not usually measured in clinical practice. In fact, their role is specifically related to the possibility of being able to interfere with peripheral levels of thyroid hormones (T4 and T3), inducing a false elevation (or decrease) of their values [142].

Another antibody has also been described in individuals with AITD. This antibody is capable of binding to colloidal antigens other than Tg (second colloidal antigen); however, the properties and function of this antibody have not been determined [143,144].

Finally, in recent years there has been a growing interest in relation to the usefulness of TPOAbs and TgAbs in individuals with Hashimoto’s encephalopathy (HE), especially since in about a third of these cases, other AIDs can coexist; additionally, in the clinical field, HE can go unnoticed (since it is a little-recognized entity). Therefore, the measurement of TPOAbs and TgAbs in individuals with cognitive impairment, dementia, stroke-like incidents, with hypothyroidism (subclinical or mild overt) can help to identify this clinical condition [145,146].

Clearly, more studies are required in this regard to be able to specify the true role of thyroid Abs in HE.

## 7. Conclusions

There have been great advances in the development of the methods used in the detection of thyroid Abs in the last few decades. Measuring these Abs helps to establish the cause of thyroid function abnormalities, as their positivity (at least for major thyroid Abs) defines the concept of thyroid autoimmunity. For instance, TPOAbs are the hallmark of HT, while TRAbs are the hallmark of GBD.

TRAbs help in the diagnostic confirmation of GBD but also in differential diagnosis with other types of hyperthyroidism; additionally, they serve as a prognostic factor in thyroid ophthalmopathy, in the risk of relapse, and in the risk of fetal/neonatal hyperthyroidism, among other functions.

Despite the usefulness that thyroid Abs provide in the study of AITD, it must be taken into account that this condition is a clinical syndrome with multiple manifestations that may overlap; therefore, the integration of clinical, imaging, and laboratory components are those that will allow a diagnosis to be made with greater certainty. The usefulness of TRAbs in the clinical approach to GBD has been demonstrated; however, in HT, the mere presence of TPOAbs and/or TgAbs are not enough to establish the diagnosis since HT is defined as a histopathological diagnosis, and although TPOAbs and/or TgAbs correlate well with histopathologic findings, their mere presence is not always a disease condition.

Other thyroid Abs (such as NISAbs, PDNAbs, and MegAbs) have not yet been shown to have a higher diagnostic performance than major thyroid antibodies; therefore, their usefulness in studying AITD has yet to be demonstrated.

## Figures and Tables

**Figure 1 antibodies-12-00048-f001:**
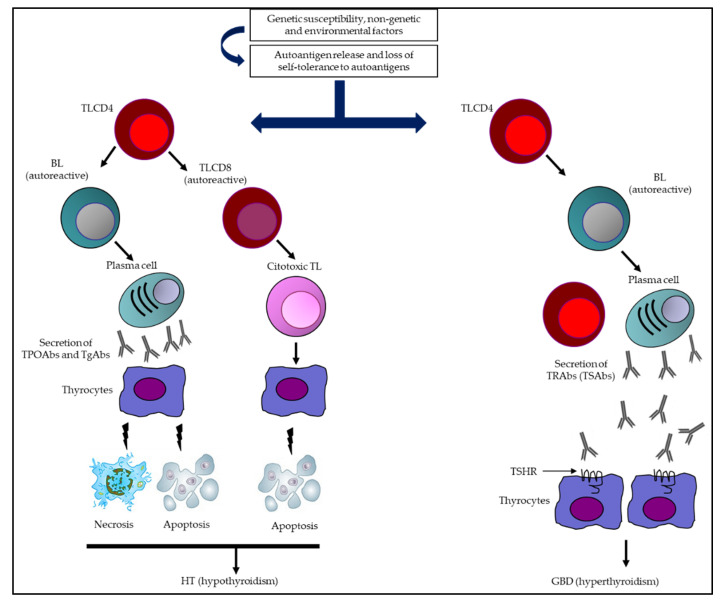
Summary of the immunological mechanisms of AITD leading to HT and GBD. Multiple genetic, epigenetic, non-genetic, and environmental factors come together in AITD, which (together with a loss of immune tolerance) are capable of inducing an immune response (humoral and cellular). In HT, the synthesis and secretion of TPOAbs and TgAbs, together with the activation of autoreactive TL, are capable of triggering destruction of thyrocytes (necrosis/apoptosis) and potentially hypothyroidism. Otherwise, in GBD, an extended humoral response predominates, with a greater capacity to secrete TRAbs (specifically TSAbs), which are the determinants of TSHR stimulation, inducing greater secretion of TH and, consequently, hyperthyroidism. Abbreviations: BL: B lymphocytes; GBD: Graves–Basedow disease; HT: Hashimoto’s thyroiditis; TgAbs: thyroglobulin antibodies; TH: thyroid hormones; TL: T lymphocytes (TL); TPOAbs: thyroid peroxidase antibodies; TRAbs: thyrotropin receptor antibodies; TSAbs: thyrotropin receptor-stimulating antibodies; TSHR: thyrotropin receptor.

**Figure 2 antibodies-12-00048-f002:**
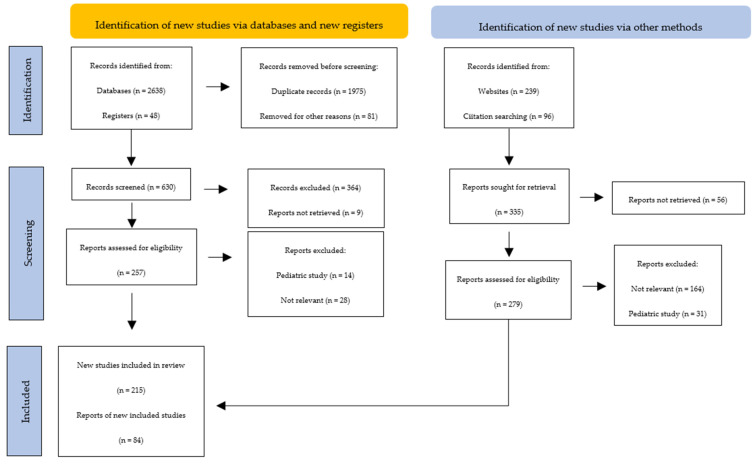
PRISMA flow diagram. Method for the selection of articles.

**Figure 3 antibodies-12-00048-f003:**
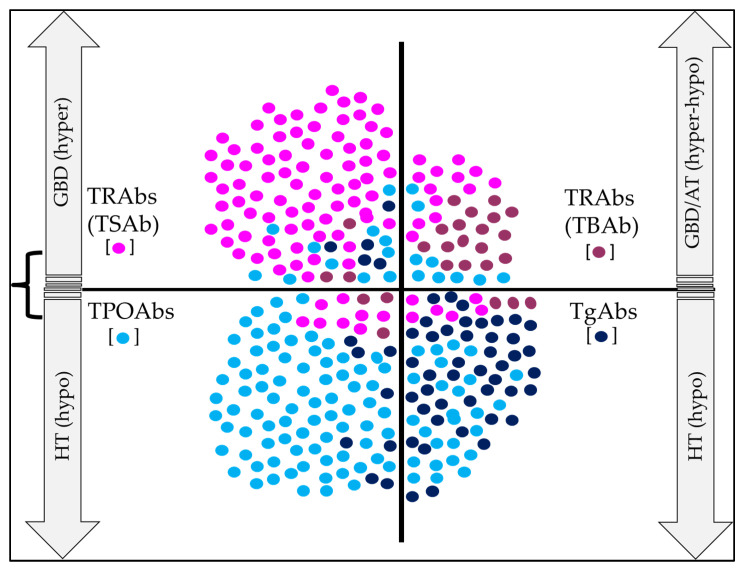
Major thyroid antibodies in the evolution of AITD. TRAbs are the hallmark of GBD, and TPOAbs are for HT; in individuals with positive TRAbs (TSAb), the risk of evolution to hyperthyroidism is significant, while in those with TBAb (and TSAb), the risk of presenting alternating hyperthyroidism and hypothyroidism (with atrophic thyroiditis) is very likely. For its part, the presence of TPOAbs and TgAbs increases the risk of presenting hypothyroidism over time. Likewise, several Abs can coexist in the same patient; therefore, some individuals who are initially classified as having HT may be reclassified over time as having GBD (and vice versa). Colored circles indicate the frequency of positivity for each Ab. Abbreviations: AT: atrophic thyroiditis; GBD: Graves–Basedow Disease; HT: Hashimoto’s thyroiditis; Hyper: hyperthyroidism; Hypo: hypothyroidism.

**Table 1 antibodies-12-00048-t001:** Molecular characteristics and prevalence of thyroid Ab positivity in subjects with AITD and in the general population References: [13,17,20,26,36,44,48,50].

Molecular Characteristics and Prevalence	Antigen					
	Tg	TSHR	TPO	PDN	NIS	Meg
Protein	Iodinated glycoprotein	G-protein-coupled receptor	Hemoprotein enzyme	Hydrophobic transmembrane glycoprotein	Membrane glycoprotein	Integral membrane protein
Amino acids	2748	743	933	780	643	4655
Molecular weight (kDa)	660	85	105–110	86	70–90	600
Thyroid concentration	++++/++++	++/++++	+++/++++	+/++++	+/++++	+/++++
Epitope localization	Predominantly central region and C-terminus	Predominantly A-subunit	Predominantly myeloperoxidase-like domain and, to lesser extent, complement control protein-like domain	Apical membrane of thyrocytes	Predominantly extramembranous Regions	Apical surface of thyrocytes
Immunogenicity	+++++/+++++	++++/+++++	+++++/+++++	+++/+++++	++/+++++	+/+++++
Chromosomal location	8q24	14q31	2p25	7q22-31	19p12-13.2	2q24-q31
Prevalence of antibody in the general population (%)	TgAbs (5–20)	TRAbs (0–3)	TPOAbs (8–30)	Unknown	Unknown	Unknown
Prevalence of antibody in autoimmune thyroiditis—HT—(%)	TgAbs (80–90)	TRAbs (10–20)	TPOAbs (90–100)	PDNAbs (variable, 9–98%)	NISAbs (7–8)	MegAbs (~50)
Prevalence of antibody in GBD (%)	TgAbs (30–60)	TRAbs (90–95)	TPOAbs (80)	PDNAbs (variable, 10–75%)	NISAbs (10–12)	MegAbs (~50)

Abbreviations: Abs: antibodies; HT: Hashimoto’s thyroiditis; Meg: megalin; NIS: sodium iodide symporter; PDN: pendrin; Tg: thyroglobulin; TPO: thyroid peroxidase; TSHR: thyrotropin receptor. The signs (+) refer to how much thyroid concentration the thyroid antigens have, and we define it on a scale that goes from 1+ to a maximum of 4+, therefore, if the thyroid concentration of Meg is appreciated (which has a very low thyroid concentration) we then place it as +/++++; likewise, if the thyroid concentration of Tg is appreciated (which has a very high thyroid concentration) then we place it as ++++/++++.

**Table 2 antibodies-12-00048-t002:** Most frequently used immunoassays for the measurement of TPOAbs and TgAbs in AITD References: [64,65,66].

Principle [kit]	CLIA [Architect (Abbott Diagnostics, USA)]	ECLIA [ECLusys (Roche Diagnostics, Germany)]	EIA [AIA-Pack (Tosoh Bioscience, Japan)]	CLEIA [Lumipulse G (Fujirebio Inc., Japan)]	CLEIA [Immulite 2000 (Siemens Healthcare Diagnostics, USA)]
Procedure	Two-step sandwich	One-step competitive	Two-step sandwich	Two-step sandwich	Two-step sandwich
TgAbs					
Assay components	Human Tg-coated Microparticles. Acridinium-labeled anti-human IgG (mouse monoclonal)	Biotinylated human Tg. Ruthenylated anti- TgAb (mouse monoclonal)	Human Tg-coated Microparticles. ALP-labeled antihuman IgG (mouse monoclonal	Human Tg-coated Microparticles. ALP-labeled antihuman IgG (mouse monoclonal). 3-(2′-spiroadamantane)-4-methoxy-4-(3″- phosphoryloxy)phenyl-1,2-dioxetane disodium salt	Human Tg-coated Microparticles. ALP-labeled antihuman IgG (mouse monoclonal). 3-(2′-spiroadamantane)-4-methoxy-4-(3″- phosphoryloxy)phenyl-1,2-dioxetane disodium salt
Cut off (IU/mL)	4.11	28	13.6	12.2	40
TPOAbs					
Assay components	Human TPO-coated microparticles. Acridinium-labeled anti-human IgG (mouse monoclonal)	Biotinylated human TPO. Ruthenylated anti- TgAb (goat polyclonal)	Human TPO-coated Microparticles. ALP-labeled antihuman IgG (mouse monoclonal	Human TPO-coated microparticles ALP-labeled antihuman IgG (mouse monoclonal). 3-(2′-spiroadamantane)-4-methoxy-4-(3″- phosphoryloxy)phenyl-1,2-dioxetane disodium salt	Human TPO-coated microparticles ALP-labeled antihuman IgG (mouse monoclonal). 3-(2′-spiroadamantane)-4-methoxy-4-(3″- phosphoryloxy)phenyl-1,2-dioxetane disodium salt
Cut off (IU/mL)	5.6	16	3.2	5.1	35

**Table 3 antibodies-12-00048-t003:** Prevalence of TgAbs and/or TPOAbs, according to the most frequently used immunoassay methods in individuals with AITD and healthy controls References: [64,67,68,69,70].

Prevalence of TgAbs and TPOAbs										
Principle [kit]	CLIA [Architect (Abbott Diagnostics, USA)]		ECLIA [ECLusys (Roche Diagnostics, Germany)]		EIA [AIA-Pack (Tosoh Bioscience, Japan)]		CLEIA [Lumipulse G (Fujirebio Inc., Japan)]		CLEIA [Immulite 2000 (Siemens Healthcare Diagnostics, USA)]	
Abs	TgAbs	TPOAbs	TgAbs	TPOAbs	TgAbs	TPOAbs	TgAbs	TPOAbs	TgAbs	TPOAbs
Hashimoto’s thyroiditis	++++ *	++	+++ *	++	+++ *	++	++++ *	++	++	++
GBD	++	++	++	++	++	++	++	++	++	++
Painless thyroiditis	++ *	+	++ *	+	++ *	+	++ *	+	+	+
Healthy controls	+ **	+/−	+	+/−	+	+/−	+ **	+/−	+/−	+/−
Prevalence of TgAbs (alone) or TPOAbs (alone)										
Abs	TgAbs	TPOAbs	TgAbs	TPOAbs	TgAbs	TPOAbs	TgAbs	TPOAbs	TgAbs	TPOAbs
Hashimoto’s thyroiditis	+ *	−	+ *	+/−	+ *	+/−	+ *	−	+	++
GBD	+	+	+/−	+	+/−	+	+	+	+/−	++ **
Painless thyroiditis	++ *	+	++ *	+/−	++ *	+/−	++ *	−	+	+/−
Healthy controls	+ **	+/−	+/−	+/−	+	+/−	+ **	+/−	+/−	+/−

Abbreviations: CLEIA: chemiluminescence enzyme immunoassay; CLIA: chemiluminescence immunoassay; ECLIA: electro-chemiluminescence immunoassays; EIA: Enzyme Immunoassay. * *p* < 0.01, ** *p* < 0.05.

**Table 4 antibodies-12-00048-t004:** Measurement methods and precision of the most commonly used TRAbs in GBD assessment in clinical practice References: [81,82,83,84].

TRAbs				
Immunoassays				ELISA
EliA Anti-TSHR (Thermo Fisher Scientific, Germany)	Elecsys (COBAS, Roche, USA)	BRAHMS TRAK Human KRYPTOR (Thermo Fisher Scientific, Germany)	IMMULITE 2000 TSI (Siemens, Healthineers, Germany)	ELISA RSR TRAb Fast (RSR Limited, United Kingdom)
Sensitivity: 96.6%	Sensitivity: 100%	Sensitivity: >98%	Sensitivity: 98.3%	Sensitivity: 85%
Specificity: 99.4%	Specificity: 95.3%	Specificity: almost 100%	Specificity: 97%	Specificity: 100%
The lower and upper limit of detection are 1.5 and 80 IU/L, respectively. Intra-assay and inter-assay variance at 3.2 U/L (positive cut-off >3.3 IU/L, negative cut-off <2.9 IU/L) is 10.6% and 11.4%, respectively.	The range is 0.8–40 IU/L. The limit of quantification is the lowest analyte concentration that can be reproducibly measured with an intermediate precision CV of ≤20%	The range is 0.27–20 IU/L (cut-off 1.8 U/L). Intra-assay variance for the range of 1.2 to 2.0 U/L is <7.0%. Inter-assay variance for the range of 1.0 to 2.0 U/L is <18%	The range is 0.10–40 IU/L (cut-off 0.55 IU/L). Intra-assay and inter-assay variance at 0.69 IU/L is 4.1% and 5.1%, respectively	The range is 1–40 IU/L (positive cut-off ≥1.0 IU/L, lower detection limit at 2 SD 0.16 IU/L). Intra-assay and inter-assay variance at 2.0 and 4.6 IU/L is reported 7.2% and 3.3%, respectively

Abbreviations: CV: coefficient of variation; SD: standard deviation.

**Table 5 antibodies-12-00048-t005:** Most commonly used measurement methods for NISAbs in AITD References: [87,88,89,90,91,92].

Principle/Kit [Ref]	Bioss Inc.’s NIS Polyclonal Antibody; USA [87]	Polyclonal Antibody to NIS, MyBioSource.com; USA [88]	Anti-NIS Polyclonal Antibody, American Research Products Inc., USA [89]	SLC5A5/NIS Monoclonal Antibody, LifeSpan BioSciences; USA [90]	NIS Antibody, Biorbyt; United Kingdom [91]	Anti-NIS Antibody, GeneTex; USA [92]
Clonality	Rabbit Polyclonal antibody	Rabbit Polyclonal antibody	Polyclonal antibody	Mouse monoclonal antibody	Human Polyclonal antibody	Rabbit Polyclonal antibody
Isotype	IgG	IgG	IgG	IgG1	IgG	IgG
Immunogen	KLH conjugated peptide, mouse NIS	Rabbit polyclonal antibody raised against NIS	Synthetic peptide from the C-terminus of rat NIS	Synthetic peptide corresponding to aa37–54 of human NIS.	KLH conjugated synthetic peptide derived from mouse NIS	KLH conjugated synthetic peptide derived between 535–608 amino acids of human NIS
Purity	Protein A purified	Affinity Chromatography	Immunogen affinity purified	Affinity purified	Affinity purified by Protein A	Protein A purified
Reactivity	Human, Rat, Pig	Human	Human, Porcine, Rat	Human	Human	Human, rat
Applications	WB, ELISA, IHC-P, IHC-F, ICC, IF	WB, ICC, IHC-P, EIA	IHC	IHC, IHC-P, WB	WB	WB, IHC-P, IHC-Fr, IHC

Abbreviations: ELISA: Enzyme-linked immunosorbent assay; ICC: Immunocytochemistry; IF: Immunofluorescence; IHC: Immunohistochemistry; IHC-F: Immunohistochemistry—fixed; IHCFr: Immunohistochemistry—frozen; IHC-P: Immunohistochemistry-Paraffin; KLH: Keyhole limpet hemocyanin; WB: Western Blot.

**Table 6 antibodies-12-00048-t006:** Most commonly used measurement methods for PDNAbs in AITD References: [93,94,95,96,97].

Principle/Kit [Ref]	Polyclonal Rabbit Anti-Human SLC26A4/Pendrin Antibody, LifeSpan BioSciences; USA [93]	SLC26A4 Antibody, MyBioSource.com; USA [94]	SLC26A4/Pendrin Monoclonal Antibody, LifeSpan BioSciences; USA [95]	Pendrin Antibody/SLC26A4, NSJ Bioreagents; USA [96]	Immunotag™ S26A4 Polyclonal Antibody, G Biosciences; USA [97]
Clonality	IgG Polyclonal	Rabbit Polyclonal	Monoclonal	Polyclonal	Polyclonal
Isotype	IgG, epitope: aa287-336	IgG	IgG2a kappa	IgG	Primary antibody
Immunogen	Synthetic peptide located between aa287-336 of human SLC26A4 (O43511, NP_000432).	Amino acids ELNDRFRHKIPVPIPIE VIVTIIATAISYGANLE KNYNAGIVKSIPRGFL	SLC26A4 (NP_000432, aa 674–754). A partial recombinant protein with GST tag. MW of the GST tag alone is 26 KDa.	Amino acids RSLRVIVKEFQRIDVNVYFASLQDYVIEKLEQ	Synthesized peptide derived from part region of human protein
**Purity**	Immunoaffinity purified	Affinity purified	Purified from ascites by Protein A	Antigen affinity purified	Affinity purified
Reactivity	Human	Human, mouse	Human	Human	Human
Applications	IHC, IHC-P, WB	WB	ELISA	WB	WB, ELISA

Abbreviations: ELISA: Enzyme-linked immunosorbent assay; IHC: Immunohistochemistry; IHC-P: Immunohistochemistry-Paraffin; WB: Western Blot.

**Table 7 antibodies-12-00048-t007:** Prevalence of positivity for NISAbs and PDNAbs in subjects with GBD, HT, and in participants without AITD References: [98,99,100,101,102,103,104].

Antibody	Participants without AITD (Controls, %)	GBD (%)	HT (%)	Ratio of Prevalence of Positivity between Subjects with AITD and Participants without AITD
Studies with NISAbs	0	84	15	4.6
	0–10	0–63	0–25.9	2.7
	0.6–3	5.6–10.7	6.9–20.8	1.5
	0	22	24	1.95
	–	38	27.5	1.75
	0	20	14	0.4
	1.8	12.3	7.5	0.8
Studies with PDNAbs	0.0	74	97.5	1.75
	0.0	9.9	7.6	4.89
	0	13	8	0.4
	5.0	11.0	4.7	0.8

Abbreviations: AITD: autoimmune thyroid disease; GBD: Graves–Basedow disease; HT: Hashimoto’s thyroiditis; NISAbs: sodium iodide symporter antibodies; PDNAbs: pendrin antibodies.

**Table 8 antibodies-12-00048-t008:** Most commonly used measurement methods for MegAbs in AITD References: [109,110].

Principle/Kit (Ref)	LRP2/Megalin Antibody, LifeSpan BioSciences; USA	Anti-Lrp2/Megalin Rabbit Monoclonal Antibody, BosterBio; USA	Rabbit Anti-Megalin/LRP2 Antibody, MyBioSource.com; USA	Human Anti-Human Megalin, Bio-Rad; USA	Mouse Anti-Human LRP2/Megalin Clone CD7D5 mAb from Cell Sciences; USA
Clonality	Polyclonal	Monoclonal	Polyclonal	Monoclonal	Monoclonal
Isotype	IgG, epitope: aa287-336	Rabbit IgG	IgG	Fab fragment	IgG1
Immunogen	Amino acids 4446–4655 of human LRP2 (NP_004516.2). FHYRRTGSLLPALPKL PSLSSLVKPSENGNGV TFRSGADLNMDIGVS GFGPETAIDRSMAMSE DFVMEMGKQPIIFENP MYSARDSAVKVVQPI QVTVSENVDNKNYGS PINPSEIVPETNPTSPA ADGTQVTKWNLFKRK SKQTTNFENPIYAQME NEQKESVAATPPPSPS LPAKPKPPSRRDPTPT YSATEDTFKDTANLVKEDSEV	A synthesized peptide derived from human Lrp2/Megalin	A synthetic peptide corresponding to the center region of the mouse LRP2/Megalin	Human Megalin (aa sequence 1024–1224)–N1 fusion protein	Purified Human Megalin
Purity	Affinity purified	Affinity-chromatography	Protein A and antigen affinity	Affinity purified	Protein G Chromatography
Reactivity	Mouse, Rat, Human	Human, Mouse, Rat	Mouse	Human	Human
Applications	IHC, IHC-P, WB	WB	IHC-P	WB	IF, IHC

Abbreviations: IF: Immunofluorescence; IHC: Immunohistochemistry; IHC-P: Immunohistochemistry-Paraffin; WB: Western Blot.

## Data Availability

No new data were created or analyzed in this study. Data sharing does not apply to this article.

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
