# Peer review of "The Usefulness of Thyroid Antibodies in the Diagnostic Approach to Autoimmune Thyroid Disease"

_2073-4468, 2023, doi:10.3390/antib12030048_

Round 1

Reviewer 1 Report

After reviewing the manuscript, the following suggestions can be made: 

In the title, the part of "thyroid antibodies" should be changed to "Anti-thyroid Antibodies" otherwise the manuscript would have no reason to exist. 

Work and improve the part of the abstract, the title does not quite match with the part that is highlighted in this section. The important part is antithyroid antibodies and their correlation with autoimmune thyroid diseases, so the core part of this relationship should be better described, briefly, but of equal importance to attract the reader through the abstract. 

Work and improve the part of the conclusion, because it is very vague and does not really specify the benefit or relevance of the results obtained. 

It should be emphasized that these clinical laboratory tests are only a part of a wide range of complementary studies to reach an etiological diagnosis.   

And it should be emphasized that they should not be, to this day, taken as definitive or diagnostic studies without the support of an integral medical approach. 

As an example, Hashimoto's disease has a histopathological diagnosis that correlates with the presence and increase of antithyroid antibodies to Thyroperoxidase, however, the simple presence of these elevated antithyroid antibodies is not always a condition for this disease.  To improve this part of the conclusion, with the purpose that it is a reflection of the good work that was carried out in the development of the manuscript. 

Author Response

Response to reviewers, article: The Usefulness of Thyroid Antibodies in the Diagnostic Approach to Autoimmune Thyroid Disease.

We take this opportunity to deeply thank the reviewers for each of the valuable comments and suggestions made on the manuscript; without a doubt they have significantly improved our work.

Next, we answer each of them point by point.

In the original document the changes are highlighted in yellow.

Reviewer 1:

  • In the title, the part of "thyroid antibodies" should be changed to "Anti-thyroid Antibodies" otherwise the manuscript would have no reason to exist. 
  • Response: Many thanks to the reviewer for the suggestion, we have made the suggested change.
  • Work and improve the part of the abstract, the title does not quite match with the part that is highlighted in this section. The important part is antithyroid antibodies and their correlation with autoimmune thyroid diseases, so the core part of this relationship should be better described, briefly, but of equal importance to attract the reader through the abstract. 
  • Response: Many thanks to the reviewer for the suggestion, we fully agree, therefore we have added a few paragraphs to the abstract, improving the aspects that the reviewer has described.
  • Work and improve the part of the conclusion, because it is very vague and does not really specify the benefit or relevance of the results obtained. It should be emphasized that these clinical laboratory tests are only a part of a wide range of complementary studies to reach an etiological diagnosis.  And it should be emphasized that they should not be, to this day, taken as definitive or diagnostic studies without the support of an integral medical approach. As an example, Hashimoto's disease has a histopathological diagnosis that correlates with the presence and increase of antithyroid antibodies to Thyroperoxidase, however, the simple presence of these elevated antithyroid antibodies is not always a condition for this disease.  To improve this part of the conclusion, with the purpose that it is a reflection of the good work that was carried out in the development of the manuscript. 
  • Response: Many thanks to the reviewer for the suggestion; we describe the usefulness and applications of the different autoantibodies in AITD in the sections: 6; 6.1; 6.2; 6.3 and 6.4; however, we fully agree with the reviewer's suggestion, therefore, we add a few paragraphs in the same direction as those suggestions.

Reviewer 2 Report

There are some points concerning the clinical utility of Thyroid Antibodies:

1. Certainly, the topic of this paper are "antibodies", but for the clinician it would be very useful to mention the association of AB positivity and the chracteristic pattern on ultrasound.

2. The authors should mention, if there are studies on the level of TRabs and the likelihood of a relapse. (f.i. studies from Wiersinga et al.)

3. Lit 128 does not mention anything on recurrence rate of GD. There is only a description on specifity and sensitivity of the assay. The authors should omit this reverence and include an appropriate one.

4. Concerning amiodarone- induced thyroidtis: 

In general the major effect of amiodarone is hyperthyroidism due to autonomy. The sentence: "Consequently, some patients......when this is not warranted"is rather confusing and should be clarified.

Author Response

Response to reviewers, article: The Usefulness of Thyroid Antibodies in the Diagnostic Approach to Autoimmune Thyroid Disease.

We take this opportunity to deeply thank the reviewers for each of the valuable comments and suggestions made on the manuscript; without a doubt they have significantly improved our work.

Next, we answer each of them point by point.

In the original document the changes are highlighted in yellow.

Reviewer 2:

  • There are some points concerning the clinical utilityof Thyroid Antibodies:

Certainly, the topic of this paper are "antibodies", but for the clinician it would be very useful to mention the association of AB positivity and the chracteristic pattern on ultrasound.

  • Response: Many thanks to the reviewer for the suggestion, certainly our article reviews and describes the usefulness of anti-thyroid antibodies in the diagnostic approach to AITD; and of course there is a very large and valuable literature regarding the indications and usefulness of thyroid ultrasonography in AITD. However, we believe that addressing this aspect in this review would escape its initial objective; therefore, we respectfully consider that this aspect should be the subject of another type of review, with other objectives and purposes.
  • The authors should mention, if there are studies on the level of TRabs and the likelihood of a relapse. (f.i. studies from Wiersinga et al.).
  • Response: Many thanks to the reviewer for the suggestion. In sections 6 and 6.3 (and in the conclusion) of the article we describe the usefulness of measuring TRAbs with respect to the risk of relapse and recurrence of the disease. References 116, 117, 119, 120, 121, 122, 123 and 124 address the utility of TRAbs and involve the excellent studies by Wiersinga et al. Therefore, the concepts outlined in these paragraphs, in addition to the results published by Wiersinga et al., also take into account other concepts described by other authors.

 Lit 128 does not mention anything on recurrence rate of GD. There is only a description on specifity and sensitivity of the assay. The authors should omit this reverence and include an appropriate one.

  • Response: Very thanks to the reviewer for the suggestion, Reference 128 is from Mathew J, et al. The objective of this study was to audit the use of the TRAb test in an outpatient endocrinology clinic and understand the real‑world utility of the test in the differential diagnosis of people presenting with suppressed TSH; in reality they determine the frequency of relapses in subjects with GBD, very surely the reviewer's doubt is in this sense, therefore, we have replaced reference 128 for a systematic review and meta-analysis that addresses this aspect.
  • Concerning amiodarone- induced thyroidtis: In general the major effect of amiodarone is hyperthyroidism due to autonomy. The sentence: "Consequently, some patients......when this is not warranted"is rather confusing and should be clarified.
  • Response: very thanks to the reviewer for the suggestion. We fully agree, therefore, we have corrected the paragraph, which reads as follows:The detection of TRAbs can also help to make the differential diagnosis in individuals with amiodarone-induced thyroiditis (AmIT), since in this scenario, the presence of TRAbs increases the positive predictive value of a type 1 AmIT; however, a minority of patients can coexist with an associated inflammatory component (type 2 AmIT), being a real challenge to be able to differentiate between both entities; inclusive, in the management decision making with ATD [125].

Reviewer 3 Report

The paper presented to me for review is a comprehensive review on the usefulness of thyroid antibodies in the diagnostic approach to ATD. This is an important clinical issue because the knowledge of the pathogenetic mechanisms of ATD continues to expand and at the same time the diagnosis poses many difficulties. 

The work is written in a good and understandable language, the PRISMA protocol is not questionable - based on it, the authors referred to the latest knowledge on the subject. 

However the work does not raise any major objections to me before accepting it for publication, I would suggest some supplementation, which may increase its value: 

1. antibody titers and their correlations with clinical manifestations in the course of neurological manifestations of ATD are repeatedly discussed. This is still a debatable issue, the data in the literature are contradictory, and it is worth mentioning this considering the following publications: PMCID: PMC7946475 and PMCID: PMC9496753

Author Response

Many thanks to the reviewer for the suggestions.

Response:

We fully agree, for this reason we have inserted a couple of paragraphs prior to the discussion with 2 references in this regard, said paragraphs were written as follows:

...Finally, in recent years there has been a growing interest in relation to the usefulness of TPOAbs and TgAbs in individuals with Hashimoto's encephalopathy (HE), especially since in about a third of these cases other AIDs can coexist; additionally, in the clinical field HE can go unnoticed (since it is a little recognized entity). Therefore, the measurement of TPOAbs and TgAbs in individuals with cognitive impairment, dementia, stroke-like incidents, with hypothyroidism (subclinical or mild overt) can help to identify this clinical condition [145,146].

Clearly more studies are required in this regard to be able to specify the true role of thyroid Abs in HE.

Round 2

Reviewer 1 Report

The suggested modifications to the document are noted. 

I consider it pertinent to indicate as a last suggestion (I apologize for not having highlighted it in the first round of revision, but I consider that it does not bring more work or complications) to indicate if the tables and images are of own authorship or taken and modified from some other text, in any of the cases the reference of where it was taken and modified should be placed, or where the support and evidence that justifies such images and tables is available. 

Author Response

Many thanks to the reviewer for the suggestions (we also hadn't noticed the details that the reviewer was able to identify).

We fully agree with the suggestions; the three figures in the article are our authorship; figure 1 for example, is preceded by the paragraphs supported in references 11 to 16 (and was designed by us).

Figure 2, is the PRISMA flow diagram (also designed by us).

Figure 3, is preceded by references 132 to 135 (it was also designed by us).

Regarding the tables, all of them were designed by us and we have placed the references that support the information contained in each of them.

Additionally, I have highlighted the new (minor) changes in the document in blue, in order to differentiate them from the changes made in the first revision.

Once again, many thanks to the reviewer, these changes will allow the article to have the necessary quality and comply with the standards and requirements of the journal.